# CRISPRing *KRAS*: A Winding Road with a Bright Future in Basic and Translational Cancer Research

**DOI:** 10.3390/cancers16020460

**Published:** 2024-01-22

**Authors:** Xian Gong, Jianting Du, Ren-Wang Peng, Chun Chen, Zhang Yang

**Affiliations:** 1Department of Thoracic Surgery, Fujian Medical University Union Hospital, 29 Xinquan Road, Fuzhou 350001, China; grox0914@163.com (X.G.); dujt1220@fjmu.edu.cn (J.D.); 2Key Laboratory of Cardio-Thoracic Surgery, Fujian Medical University, Fuzhou 350001, China; 3Division of General Thoracic Surgery, Department of BioMedical Research (DBMR), Inselspital, Bern University Hospital, University of Bern, Murtenstrasse 28, 3008 Bern, Switzerland; renwang.peng@insel.ch

**Keywords:** KRAS, CRISPR, gene editing, resistance, synthetic lethal

## Abstract

**Simple Summary:**

The development of proprietary drugs targeting KRAS-mutant tumors has historically been a formidable challenge. This difficulty stems from the high affinity of RAS proteins for GTP and the lack of a hydrophobic “pocket” conducive to drug binding. However, the emergence of CRISPR technology, a groundbreaking gene-editing tool, has revolutionized tumor studies, particularly those focusing on KRAS mutations. This article offers a review of both fundamental and translational research leveraging the CRISPR system in the context of KRAS-mutant cancer. It encapsulates recent strides made in understanding KRAS biology’s mechanistic nuances, shedding light on pivotal themes such as drug resistance, anti-tumor immune responses, epigenetic regulation, and the exploitation of synthetic lethality by mutant KRAS. In conclusion, the article touches upon the current limitations of employing CRISPR technology in KRAS-related research, while also suggesting avenues for future refinement and optimization in this dynamic field.

**Abstract:**

Once considered “undruggable” due to the strong affinity of RAS proteins for GTP and the structural lack of a hydrophobic “pocket” for drug binding, the development of proprietary therapies for KRAS-mutant tumors has long been a challenging area of research. CRISPR technology, the most successful gene-editing tool to date, is increasingly being utilized in cancer research. Here, we provide a comprehensive review of the application of the CRISPR system in basic and translational research in KRAS-mutant cancer, summarizing recent advances in the mechanistic understanding of KRAS biology and the underlying principles of drug resistance, anti-tumor immunity, epigenetic regulatory networks, and synthetic lethality co-opted by mutant KRAS.

## 1. Introduction

Kirsten rat sarcoma viral oncogene homolog (KRAS) mutations are the most common genetic alterations in human tumors [1], predominantly found in pancreatic ductal adenocarcinoma (PDAC), colorectal cancer (CRC), and lung adenocarcinoma (LUAC) [2], affecting over 95%, 40%, and 30% of the patients, respectively [3]. The *KRAS* gene encodes two distinct proteins, KRAS 4A and KRAS 4B, through selective exon splicing, with the latter being the primary form of the KRAS protein in human tumors [4]. Functionally, KRAS is a member of the guanosine triphosphate (GTP) binding protein family [5], also known as the RAS superfamily or RAS-like GTPases, and acts as a “switch” between inactive guanosine diphosphate (GDP) and active guanosine triphosphate (GTP), which transduces upstream signaling to various downstream pathways [6].

KRAS activity is dynamically regulated by GTPase-activating proteins (GAPs) and guanine nucleotide exchange factors (GEFs). While GAPs facilitate the conversion of GTP to GDP, GEFs trigger the exchange of GDP for GTP. SOS1/2 (son of sevenless homolog 1/2) genes are primary GEFs of the RAS that activate the RAS by interacting with the Src homology 3 (SH3) domain of growth factor receptor-bound protein 2 (GRB2) [7] and Src homology phosphatase 2 (SHP2) [8]. They are primarily recruited to the plasma membrane by binding to specific membrane-associated proteins and undergo a conformational change that relieves their autoinhibition, allowing them to interact with RAS proteins. GTP-bound KRAS then stimulates a variety of signaling pathways, including the mitogen-activated protein kinase (MAPK) pathway [9], PI3K/AKT/mTOR pathway, RAS-like protein (RAL) guanine nucleotide dissociation stimulator (RalGDS), and TIAM Rac1 Associated GEF 1 (TIAM1, a guanine nucleotide exchange factor that facilitates the exchange of GDP for GTP on the Rac1 protein) [9,10].

Mutations in the KRAS gene, such as the G12C substitution, interfere with the RAS–GAP interaction, diminish GTPase activity, maintain RAS in an activated state, and lead to the hyperactivation of downstream signaling pathways. For many years, KRAS mutations have been considered “undruggable”, primarily due to the strong affinity of RAS proteins for GTP, which, when reaching picomolar levels, makes it difficult to find competitive inhibitors with comparable affinity [2,11,12]. In addition, the surface of KRAS lacks an ideal hydrophobic “pocket” structure for drug binding [13,14]. As a gain-of-function mutation, novel inhibitors targeting KRAS G12C have recently been developed and have entered clinical trials, the most impactful being AMG510 (Sotorasib) and MRTX849 (Adagrasib), which have received accelerated approval from the US Food and Drug Administration (FDA) for the treatment of non-small cell lung cancer (NSCLC) with the KRAS-G12C mutation [15,16,17]. They bind irreversibly and selectively to the switch II pocket within the mutant KRAS, locking it in the inactive GDP binding state. In addition, some research also indicates that directly targeting the degradation of the KRAS protein is a potential treatment strategy that can induce the regression of tumors expressing mutant KRAS in mouse models [18].

The CRISPR system is an adaptive immune system discovered in prokaryotes [19]. In 2013, the CRISPR/Cas system was used to precisely edit genes in mammalian cells [20] and has since become a highly efficient, low-cost gene-editing technology that undoubtedly holds great promise for cancer research (Figure 1). In particular, CRISPR/Cas technology has led to significant scientific advances in the study and treatment of KRAS-driven cancers, broadening our understanding of mechanistic insights into KRAS-driven tumor initiation, progression, and resistance to therapy, as well as synthetic lethal interactions with mutant KRAS.

## 2. Application of CRISPR/Cas Technology to Basic and Translational Cancer Research on KRAS

### 2.1. Mechanisms of KRAS-Dependent Oncogenesis

Oncogenic mutations in KRAS disrupt its interaction with GAP, as seen in the G12 (G12D, G12C, G12V, etc.) and G13 (G13D, G13V, etc.) mutations, or inhibit its GTPase activity, as seen in the Q61 (Q61K, Q61R, etc.) mutation [21]. These modifications result in the KRAS protein persisting in a GTP-bound state, thereby continuously signaling to a number of downstream effector pathways, including but not limited to the MAPK pathway, PI3K/AKT/mTOR pathway, RalGDS, and TIAM1. Sustained activation of these pathways facilitates the oncogenic transformation of cells and confers growth, survival, and metastatic advantages to cancer cells (Figure 2) [22]. Comprehensive sequencing studies have revealed numerous genetic alterations that co-occur with KRAS mutations. However, the function of these co-mutated genes in cancer development and progression is not fully understood. There has been extensive research aimed at understanding the underlying mechanisms; the ease of use of CRISPR gene-editing technology has facilitated the development of various mouse models (Appendix A) and has advanced our understanding of the mechanisms of co-occurring mutations in KRAS-driven oncogenesis [23,24,25,26].

The TP53 gene, which encodes the p53 protein in both humans and mice, is a key tumor suppressor gene [27] that is commonly co-mutated with KRAS. The transition from normal colon epithelial cells to colon cancer cells requires several molecular changes. First, loss of the adenomatous polyposis coli (APC) gene triggers precancerous lesions. Subsequently, mutations in genes such as KRAS, SMAD4, and TP53 facilitate progression to a cell phenotype characterized by invasiveness and metastasis [28]. In rodent models, the introduction of a pair of driver pathway mutations has the potential to induce colorectal cancer. However, the number of driver pathway alterations in human colorectal cancer is highly variable, with some cases showing no pathway alterations or a single pathway alteration [29]. Further research is needed to elucidate the role of driver pathway alterations in human colorectal cancer. In 2015, Matano [30] and colleagues used CRISPR technology to introduce mutations in the tumor suppressor genes APC, SMAD4, and TP53 and the oncogenes KRAS and/or PIK3CA into human intestinal epithelial organoids. They cultured these organoids in conditions simulating the intestinal environment and found that tumors formed under the renal capsule in mice after implantation. Interestingly, the metastasis of these cells to the liver and spleen was less aggressive than that of human adenoma organoids derived from unstable chromosomes. This suggests that while driver pathway mutations may enable stem cells to survive in an unfavorable tumor microenvironment, other factors also influence invasive behavior, such as distant metastasis. As an anti-cancer gene therapy, current applications of novel CRISPR technologies, such as base editing and prime editing, can correct the genome without deleting the genes and can be used to edit TP53 missense mutations [31].

Romero et al. [32] used the CRISPR/Cas9 system to study the role of Kelch-like ECH-associated protein 1 (KEAP1) in KRAS-mutant NSCLC progression and found that loss of KEAP1 leads to the overactivation of nuclear factor erythroid-2-like 2 (NFE2L2), which encodes the transcription factor NRF2 and promotes KRAS-driven lung adenocarcinoma in mice. Using gene screening and metabolomics analysis in conjunction with the CRISPR/Cas9 system, Romero et al. showed that cancers with KEAP1 or NRF2 mutations rely on increased glutamine metabolism, providing a rationale for glutaminase inhibitors in the treatment of patients with KRAS/KEAP1 or KRAS/NRF2 mutant lung cancer.

Liver kinase B1 (LKB1, also known as STK11) is another tumor suppressor gene [33] that is frequently co-mutated with KRAS in NSCLC (Figure 3) [34,35]. LKB1 is a serine/threonine kinase that regulates cell metabolism, cell polarity, and growth through the 14 AMP-activated protein-kinase-related kinases (AMPKRs), including salt-inducible kinase (SIK; including SIK1, SIK2, and SIK3) [36]. Hollstein et al. [37] used CRISPR technology to study the role of SIK in Kras-G12D mice and found that loss of SIK1 and SIK3 redundantly mediated the tumor-suppressor activity of LKB1 and accelerated tumor growth in KRAS-driven lung cancer.

Phosphatase and tensin homolog (PTEN) negatively regulates the mTOR pathway [38]. A study using CRISPR/Cas9 technology to compare the effects of STK11 and PTEN on lung cancer development confirmed that mutations in STK11, but not PTEN, promote the progression of KrasG12D mutation lung adenocarcinomas [39].

Cheng R et al. [40] reported in 2023 a new protein encoded by the long intergenic non-protein coding RNA 00673 (LINC00673). Due to its significant association with the RAS signaling pathway, it was named RASON. RASON positively regulates the oncogenic RAS signaling pathway and it has been shown that using CRISPR to knock out RASON in mouse embryonic fibroblasts can suppress the tumorigenic transformation associated with KRAS mutations.

SOX9 is a transcription factor of the high mobility group (HMG) box family that is often dysregulated in cancer and has been reported as both an oncogene [41,42] and a tumor suppressor gene [43,44]. Using CRISPR/Cas9 and Cre-LoxP gene knockout in a mouse model of Kras-G12D lung adenocarcinoma, Zhong et al. [45] found that loss of SOX9 significantly reduced the incidence, burden, and progression of mutant KRAS lung tumors by promoting anti-tumor immunity, suggesting a role for SOX9 in regulating the tumor microenvironment.

Nuclear receptor binding SET domain protein 2 (NSD2) is a key epigenetic enzyme that generates the classic histone modification H3K36me2. In most cultured cell types, NSD2 physiologically produces large amounts of H3K36me2 and the disruption of H3K36me2 caused by chromosomal translocations and functional gain-of-function mutations is etiologically associated with several hematological malignancies [46]. Sengupta et al. [47] developed a CRISPR-based system to test the gene function in KRAS mutation lung adenocarcinoma and found that NSD2 deficiency could significantly inhibit tumor progression. NSD2 knockdown could also inhibit the tumor growth in xenografts derived from patients with primary lung adenocarcinoma. The experimental results showed that the NSD2 depletion combined with the MEK1/2 inhibition treatment regimen can significantly inhibit KRAS-mutant lung adenocarcinoma tumors. It was further reported that the NSD2–h3k36me2 axis plays a key epigenetic role in maintaining carcinogenic signaling pathways.

### 2.2. Mechanisms of Resistance to KRAS Pathway Inhibition

The MAP (RAF-MEK-ERK) kinase cascade is one of the key effectors downstream of KRAS and, as such, is one of the most focused targets for blocking KRAS oncogenic activity [9,48]. Several potent and selective inhibitors of the MAPK pathway have been developed [49,50]; however, their clinical efficacy as monotherapy is limited due to drug resistance and toxicity to normal cells [51,52]. One of the resistance mechanisms is mediated by the treatment-induced loss of ERK-dependent negative feedback against the upstream receptor tyrosine kinase (RTK) signaling pathways (such as receptors for ERBB, PDGF, VEGF, FGF, and so on), which, in turn, reactivates the MAPK signaling [53,54]. Klomp et al. [55] performed genetic screening and identified the ataxia-telangiectasia-mutated-and-Rad3-related kinase (ATR, Serine/Threonine Kinase-checkpoint kinase 1 (CHK1, Serine/Threonine Kinase)) DNA damage repair (DDR) pathway that regulates extracellular signal-regulated kinase (ERK) inhibitor sensitivity in KRAS-mutant PDAC. ATR is a member of the PIKK family, with a structure similar to that of ATM and DNA-PKcs. CHK1 is the primary downstream target of ATR and is involved in DNA damage repair. CHK1 inhibition downregulates the transcription factor MYC (myelocytomatosis viral oncogene homolog) but also activates ERK and AMP kinase and induces autophagy, providing a mechanistic basis for co-targeting CHK1 and ERK and/or inhibiting autophagy for the treatment of PDAC, the most common type of pancreatic malignancy characterized by invasive growth and early metastasis. The researchers used CRISPR-based functional studies to better understand how suppression of CHK1-induced ERK activation contributes to the progression of PDAC. They identified a DNA-binding protein, Replication Timing Regulatory Factor 1 (RIF1), which is a critical element of non-homologous end joining (NHEJ) downstream of ERK. In particular, inhibition of CHK1 can increase the sensitivity of PDAC cells to further CHK1 suppression.

Adaptive activation of bypass pathways has emerged as a critical mechanism of resistance to KRAS signaling inhibition [56]. Lou et al. [57] performed CRISPR functional genomics in KRAS-G12C-mutant lung cancer and pancreatic cancer cells and identified genes necessary for cancer cell survival when KRAS signaling is blocked by KRAS-G12C inhibitors, some of which have been termed “collateral dependencies” (CDs). These include GEFs (such as SOS1), their regulators (FGFR1, EGFR, SHP2, and CRKL), and various cell growth and survival pathways (such as CCND1, CDK4, ITGA7, ITGAV, FOSL1, and PI3K/AKT/mTOR); when these genes were knocked out, the sensitivity of the cells to KRAS G12C inhibitors was increased. Based on these findings, Lou et al. suggested that the best approach to targeting KRAS-mutant tumors should consider the upstream and downstream pathways of KRAS and the CDs.

### 2.3. Epigenetic Regulatory Networks in the Development of KRAS-Mutant Tumors

Epigenetics plays a critical role in tumor biology [58] and CRISPR/Cas technology has shown great potential for studying epigenetics. In 2018, Wu et al. [59] used CRISPR/Cas9 to explore tumor suppressor genes in a Kras-G12D mouse model and found that the absence of the epigenetic regulatory factor Utx significantly accelerated lung cancer progression. This pro-tumor effect of Utx knockout in vivo primarily occurs through upregulation of an enhancer of zeste 2 polycomb repressive complex 2 subunit (EZH2), a histone-lysine N-methyltransferase, and, subsequently, histone H3 lysine 27 (H3K27) me3 levels, suggesting that UTx is a crucial epigenetic regulatory factor in lung tumor development.

The conventional CRISPR/Cas9 system introduces irreversible DNA damage mutations into the genome that are toxic due to double-strand breaks (DSBs) [60,61]. Chromatin immunoprecipitation (ChIP) experiments confirmed acetylation of the KRAS promoter histone modified by dCas9-HDAC1. Liu et al. [62] designed a catalytically inactive Cas9 protein (dCas9)- histone deacetylase 1 (HDAC1) fusion protein. By using dCas9 as a DNA binding agent and fusing it with the transcription inhibitor HDAC1, the KRAS gene is epigenetically silenced without DNA damage or double-strand breaks. Additionally, dCAS9-HDAC1 can effectively silence KRAS, significantly inhibit cell growth and soft agar colony formation, and induce cell death.

Moreover, cancer cells with KRAS mutations appear to be more dependent on NPM1 expression. Li et al. [63] carried out a whole-epigenome CRISPR knockout in vitro and in vivo to screen for epigenetic regulatory factors that could serve as novel targets for NSCLC treatment. They identified the histone chaperone nucleophosmin 1 (NPM1) as a potential treatment target. In vitro and in vivo experiments showed that silencing the NPM1 gene significantly inhibits tumor progression.

### 2.4. Crosstalk between KRAS Signaling and Anti-Cancer Immunity

Immunotherapy has emerged as a new option in cancer treatment and has shown promising results in a variety of cancers [64,65]. KRAS mutations have been shown to upregulate the immune checkpoint programmed cell death protein 1 (PD-1)/programmed death ligand 1 (PD-L1) [66,67] and promote immune escape in KRAS-driven tumors. The advent of CRISPR/Cas technology has allowed a more in-depth exploration of the mechanisms of immune resistance in KRAS-mutant tumors.

PD-L1 plays a key role in regulating the body’s immune response. It is found on the surface of various cells, including immune cells and certain cancer cells. The expression level of PD-L1 can be used to select patients who are more likely to respond to PD-L1/PD-1 inhibitors. In certain cancers, high levels of PD-L1 expression on tumor cells are associated with increased response rates to PD-L1/PD-1 inhibitors. Huang et al. [68] discovered that the tumor suppressor tuberous sclerosis complex subunit 1/2 (TSC1/2) effectively regulated PD-L1 expression in vitro and that its knockout enhanced PD-L1 transcription and membrane expression, thereby sensitizing cancer cells to PD-1 treatment in a murine Kras^G12D^/Trp53^−/−^ lung cancer model. In a syngeneic mouse model, TSC2-knockout tumors responded significantly to PD-1 treatment while TSC2 wild-type tumors did not. NSCLC patients with TSC1 or TSC2 mutations who received an immune checkpoint blockade (ICB) showed durable clinical benefits and prolonged survival.

Falcomatà et al. [69] integrated single-cell RNA sequencing, CRISPR screening, and immune phenotyping techniques and showed synergistic effects between the MEK inhibitor trametinib and the multikinase inhibitor nintedanib in KRAS mutant tumors. This drug combination induced cell cycle arrest and cell death via the cancer cell secretome and remodeling of immune-suppressive cells.

Dervovic et al. [70] performed a CRISPR/Cas9 screen in a Kras-G12D mouse model of lung cancer and identified the cancer-testis antigen a disintegrin and metalloprotease 2 (ADAM2) as an immune modulator. The authors found that ADAM2 expression limited the expression of the immune checkpoint inhibitors PD-L1, LAG3, TIGIT, and TIM3 in the tumor microenvironment and provided an explanation for why adoptively transferred cytotoxic T cells exhibited greater cytotoxic effects in tumors overexpressing ADAM2 in vitro.

There is increasing evidence that epigenetic factors play a role in regulating the tumor immune microenvironment (TME) and modulating anti-tumor immune responses. DNA methyltransferases (DNMTs), histone deacetylases (HDACs), EZH2, bromodomain protein 4 (BRD4), and lysine-specific histone demethylase 1A (LSD1), which play important roles in tumor biology, have also been shown to regulate anti-tumor immunity as well [71,72]. However, how epigenetic regulators modulate cancer immunotherapy remains underexplored. Li et al. [73] conducted an in vivo CRISPR screening using a sgRNA library targeting epigenetic factors in the Kras^G12D^/p53^−/−^(KP) mouse model of lung adenocarcinoma and found that the absence of the histone chaperone antisilencing function 1A (ASF1A) sensitizes the tumor to anti-PD-1 treatment, providing the basis for a novel combination treatment of ASF1A inhibition and anti-PD-1 immunotherapy.

### 2.5. KRAS Synthetic Lethal Interactions

In cancer, “synthetic lethality” refers to pairs of genes, where the mutation or inactivation of one gene produces oncogenic stress, forcing cells to adapt to this stress by relying more on another pathway. When this pathway is inhibited, it results in cell death [74]. In the treatment of KRAS mutant cancers, the key to exploiting synthetic lethality is to find stable mechanisms of synthetic lethal gene interactions; CRISPR/Cas technology has advanced the study of KRAS synthetic lethal interactions, partly due to the advantage of low false positive rates compared to other genetic approaches [75,76].

In 2018, Šuštić et al. [77] discovered that loss of the endoplasmic reticulum (ER) stress sensor inositol-requiring enzyme 1 (IRE1) is synthetic lethal in RAS-mutant yeast cells. However, as a human homologous gene of IRE1, knockout of ERN1 (endoplasmic reticulum to nucleus signaling 1) in KRAS-mutant colorectal cancer cells does not affect cell growth but does sensitize the cells to MEK inhibition. They further identified JUN-terminal kinase (JNK)/MAPK8 or TAK1/MAP3K7 as transducing the signal from ERN1 to JUN; the simultaneous inhibition of this pathway and MEK results in synthetic lethality in KRAS-mutant human colon cancer cells.

Kelly et al. [78] generated a protein–protein interaction network of RAS interactors using affinity purification mass spectrometry. Using this network, the authors constructed a CRISPR double-knockout library of RAS interactor genes and identified a synthetic lethal interaction of Rap1 GTPase-GDP dissociation stimulator 1 (RAP1GDS1) with ras homolog family member A (RHOA) in KRAS-mutant lung adenocarcinoma.

Farnesyl thiosalicylic acid (FTS) is a RAS inhibitor with limited efficacy in PADC. Du et al. [79] conducted a whole-genome CRISPR screen on murine PDAC cell lines treated with FTS and identified several ER-associated protein degradation (ERAD) pathway genes whose inhibition shows a synergistic effect with FTS by inducing apoptosis in KRAS-mutant pancreatic cancer cells.

The SHP2 phosphatase promotes the activation of KRAS and the downstream MAPK pathway [80]. Li et al. [81] performed whole-genome CRISPR screening on SHP2-inhibited and -uninhibited KRAS-mutant gastroesophageal adenocarcinoma (GEA, a type of cancer that originates from the glandular cells in the stomach or lower part of the esophagus, also known as adenocarcinoma of the gastroesophageal junction) cells and identified the MAPK pathway and the upstream RTK pathway, the inhibition of which enhances the efficacy of SHP2 inhibition in KRAS-mutant GEA.

The anti-apoptotic protein B-cell lymphoma extra large (BCL-XL) is overexpressed in patients with KRAS/BRAF-mutant colorectal cancer. Jung et al. [82] conducted a whole-genome CRISPR/Cas9 screen and found that the resistance to the BCL-XL inhibitor ABT-263 in KRAS-mutant colorectal cancer cells is associated with activation of the WNT pathway. In KRAS/BRAF mutant cells, genetic and drug inhibition of the WNT signaling pathway (using β-catenin shRNA or the Traf2- and Nck-interacting protein kinase inhibitor NCB-0846, respectively) can enhance the tumoricidal effect of ABT-263.

WNT inhibition transcriptionally inhibits anti-apoptotic myeloid cell leukemia-1 (MCL1), which is achieved by functional inhibition of the β-catenin complex at the MCL1 promoter. The combination of ABT-263 and NCB-0846 synergistically inhibited KRAS-mutant tumor growth in patient-derived xenograft (PDX) models.

Through integrated functional studies (knockdown by siRNAs and shRNAs, knockout with CRISPR/Cas9), we showed in 2022 that co-targeting nucleolar protein 5A (NOP56) and mTOR leads to synthetic lethality in KRAS-mutant NSCLC cells in vitro and in vivo [83].

Goodwin et al. [84] conducted CRISPR knockout screening in pancreatic cancer cells harboring KRAS mutations and identified a number of functionally diverse genes whose inhibition can enhance the growth-inhibiting effect of CDK4/6 inhibitors. These genes were enriched in several signaling pathways, including cell-cycle regulation, PI3K–AKT–mTOR, steroid receptor coactivator (SRC) family kinases, HDAC proteins, autophagy, chromosomal regulation and maintenance, and DNA damage repair. In particular, CDK4/6 inhibitors in combination with MAPK inhibitors synergistically block the compensatory upregulation of ERK, PI3K, and anti-apoptotic signaling pathways, thereby synergistically inhibiting the growth in PDAC cell lines and organoids.

In addition to the above research, numerous studies have investigated the synthetic lethality of KRAS using CRISPR/Cas technology, as summarized in Table 1.

## 3. Conclusions

Despite the high incidence of KRAS mutations in pancreatic, colorectal, and lung cancers, there are still few effective treatments for KRAS-driven cancers. For example, despite recent advances in the treatment of NSCLC with immune checkpoint inhibitors of PD1 and PD-L1, they have not shown a more pronounced effect in KRAS-mutant NSCLC compared to other NSCLC. Covalent KRAS inhibitors have been introduced into clinical practice but only for KRAS-G12C mutation alleles. In addition, targeting downstream effectors of KRAS has been extensively explored; however, their pleiotropy, complex interactions between individual signaling cascades, and toxicity caused by sustained inhibition of multiple KRAS effector pathways have hindered the translational potential of this strategy. Therefore, there remains an urgent and unmet need for a better understanding of KRAS biology and innovative treatment strategies tailored to KRAS-mutant cancers.

The CRISPR/Cas system has demonstrated its potential for basic and translational cancer research, particularly in the mechanistic understanding of KRAS biology and the development of KRAS-mutant cancers. Relevant aspects range from KRAS-dependent oncogenesis to mechanisms of resistance to KRAS inhibitors, epigenetics in KRAS mutant tumors, KRAS signaling in cancer immunity, and KRAS synthetic lethality. Some studies have successfully used CRISPR/Cas to generate CAR T cells and have progressed to the stage of clinical trials. Notably, the US FDA approved the first therapy based on CRISPR-Cas9 technology for the treatment of sickle cell anemia on 8 December 2023. The advent of the CRISPR/Cas technology, which simplifies gene editing and increases efficiency compared to RNA interference, has opened up unprecedented possibilities in cancer research, particularly in investigating the molecular mechanisms underlying the KRAS-driven tumor initiation, progression, and response to anti-cancer therapies.

Nevertheless, the current state of the CRISPR/Cas system is not without its challenges. The main problem is its off-target effect, which compromises the specificity of gene editing. In addition, the system lacks an optimal delivery method and the introduction of irreversible DNA damage mutations leading to DSBs hinders its application. In response to these issues, many researchers are actively working to refine and optimize the CRISPR/Cas system.

## Figures and Tables

**Figure 1 cancers-16-00460-f001:**
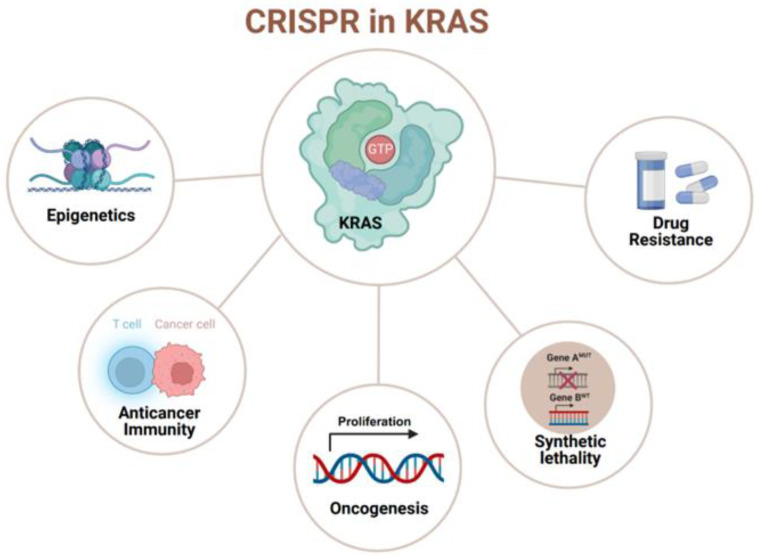
The application of CRISPR technology to basic and translational cancer research in KRAS-driven cancers. (created with BioRender.com, accessed on 29 December 2023).

**Figure 2 cancers-16-00460-f002:**
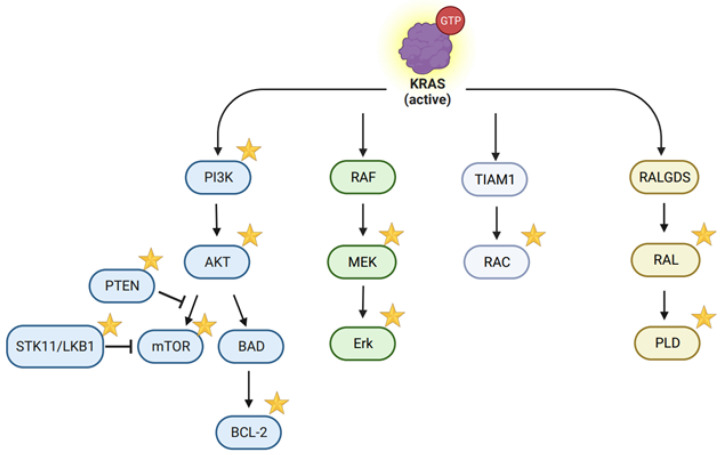
The major signaling pathways downstream of KRAS; 
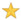
: it has been targeted or studied by the CRISPR/Cas system. (created with BioRender.com, accessed on 29 December 2023).

**Figure 3 cancers-16-00460-f003:**
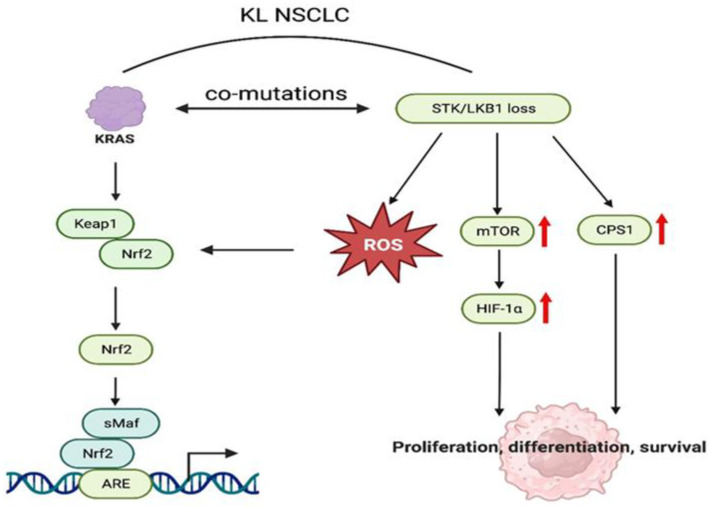
Interaction between KEAP1 and LKB1 in KRAS-mutant NSCLC. KL, KARS, and LKB1 co-mutation; sMaf, small musculoaponeurotic fibrosarcoma; ARE, antioxidant response element; HIF-1α, hypoxia-inducible factor 1 α; CPS1, carbamoyl phosphate synthetase 1; ROS, reactive oxygen species. (created with BioRender.com, accessed on 29 October 2023).

**Table 1 cancers-16-00460-t001:** KRAS synthetic lethalities identified by CRISPR/Cas technology.

Cell Lines	CRISPR Library	Synthetic Lethal Genes or Pathways	Drug Inhibition	Types of Inhibition	Reference
HCT11, SW620 CRC119, CRC240 (CRC)	Subgenomic	MEK/ERK; SRC; BCL-XL	VX-11e Dasatinib WEHI-539	KRAS effectors/tumor stress-response pathways	[85]
NB4; PL-21; SKM-1 (myeloid leukemia)	Genome-wide	RCE1; I CMT; RAF1; SHOC2; PREX1	N/A	KRAS effectors	[86]
HCT116 (colorectal cancer)	Genome-scale	SUCLA2; NADK; KHK; SNRPC; POP5; SF3B2; NF2; RALGAPB; INO80C	N/A	Tumor stress response pathways	[87]
MOR (NSCLC)	“druggable genome”	MAPK7	cobimetinib; XMD17-109	KRAS effectors	[88]
PDX366 (PDAC)	“Nuclear” sgRNA library	CENPE; RRM1	trametinib GSK923295 COH29	KRAS effectors	[89]
ERN1 knockout (KO) LoVo, HCT-116, SW480, DLD1 (colon cancer)	The human GeCKO v. 2	DUSP4; STK40, DET1; COP1; CBFB; RUNX2; JNK signaling	selumetinib (AZD6244),trametinib (GSK1120212); SR-3306;	KRAS effectors	[77]
Pancreatic and lung cancer cells	The Avana-4 barcoded sgRNA library	SHOC2/MEK	trametinib	Tumor stress-response pathways	[90]
MIA-PACA2	The Avana4 lentiviral library	PI3K/ERBB-family RTK signaling/mTOR	BYL719; pelitinib; AZD2014	KRAS effectors	[91]
H23 (NSCLC)	The genome-wide CRISPR library	IGF1R signaling pathways; CPD	N/A	KRAS effectors	[92]
PDX366 (PDAC)	8031 sgRNA–containing library	PRMT5	EPZ015666 and EPZ015938	Tumor stress-response pathways	[93]
A549; H23; H2009 (lung cancer),	The large DrugTarget-CDKO library	RAP1GDS1/RHOA	N/A	KRAS effectors/Tumor stress-response pathways	[78]
4292; PANC-1; MIA PaCa-2 (PDAC)	Pooled mouse CRISPR lentiviral library	ERAD pathway	eeyarestatin I (EerI)	Tumor stress-response pathways	[79]
SW620; HCT116 (colorectal cancer)	Genome-wide CRISPR/Cas9 Knockout (MAGeCK) algorithm	WNT signaling/BCL-2 family genes	NCB-0846; Bcl-2i: ABT-263	KRAS effectors	[82]
Pa16C (pancreatic cancer)	Not mentioned	SRC; ERK	KX2-391 (tirbanibulin)	Tumor stress-response pathways	[94]
HCT116, SW480, LS174T (colon cancer)	Genome-wide	GRB7	N/A	KRAS effectors	[95]
H358, H460, A549, PF563, PF139 (lung cancer), MIA-PaCa, HPAF-II (pancreatic cancer) HCT-116, DLD-1 (colon cancer)	NOP56 Knockout	NOP56 and mTOR	rapamycin; shNOP56	KRAS effectors/tumor stress-response pathways	[83]
Pa01C; Pa02C; Pa03C; Pa04C; Pa14C; Pa16C (PDAC)	“Druggable genome”	ERK; CDK2/4/6	PF-06873600; SCH772984	KRAS effectors	[84]
KE-39, HUG1-N (gastric cancer)	Genome-wide	SHP2/downstream MAPK pathways	SHP099; Ribociclib	KRAS effectors	[81]

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
