# Peer review of "CRISPRing KRAS: A Winding Road with a Bright Future in Basic and Translational Cancer Research"

_cancers, 2024, doi:10.3390/cancers16020460_

Round 1
Reviewer 1 Report (Previous Reviewer 4)
Comments and Suggestions for Authors
The Authors addressed most of the concerens/comments and revised the manuscript accordingly. The manuscript has been significantly improved and warrants publication. Very minor Enslish editing is still needed.
Author Response
We thank the reviewer for their constructive comments and edited the manuscript in English.
Reviewer 2 Report (Previous Reviewer 3)
Comments and Suggestions for Authors
Corrections have been made. The manuscript can be accepted now.
With regards to TP53 mutations – authors are kindly encouraged to cite the following article that describes the potential use of CRISPR/Cas9 for therapeutic editing of the TP53 gene. DOI: 10.3390/genes11060704
Author Response
Reply: We thank the reviewer for the highly positive assessment of our work,and We have cited the article in the manuscript as required. See ref. 42.
Reviewer 3 Report (Previous Reviewer 2)
Comments and Suggestions for Authors
The manuscript improved significantly, the requested figures, text passages and references have been added. I now recommend its publication.
Comments on the Quality of English Language
The text still contains typos and errors. It should be thoroughly proofread by a person who is proficient in English as well as in the scientific field.
Author Response
We thank the reviewer for the positive assessment of our work and we have re-edited the manuscript in English.
Reviewer 4 Report (Previous Reviewer 1)
Comments and Suggestions for Authors
Dear Xian Gong,Jian-Ting Du,Ren-Wang Peng, Chun Chen and Zhang Yang
Thank you very much for the thorough response to my previous comments.
The manuscript entitled “CRISPRing KRAS: a winding road with a bright future in basic and translational cancer research” reads much better now.
I have however still a few comments:
L46 please specify in how SOS1/2 interacts with RAS
L50 please introduce TIAM Rac1 Associated GEF 1 (TIAM1)
L58 please include that the G12C mutation is a gain-of-function mutation
L59 please introduce l AMG510 (Sotorasib) and MRTX849 (Adagrasib)
L79-80 please include the full mutation ie G12C instead of only G12, the same applies to G13 and Q61
L114 please include the full name of NSCLC
L119 metabolism may work better than degradation
L127 LKB1 kinase
L130 STK11 needs an introduction – please explain here the link to KRAS
Please refer to the figures in the text as appropriate. It may work better to move Fig 3 to the beginning of the review after Fig 1.
L139 please introduce Ras-on
L147 are these KRAS mutated lung tumors?
L150-161, the link to KRAS is not evident in this section, please explain
L163 MAP (RAF-MEK-ERK) kinases
L168 please introduce RTK signaling
L170-173, this section is difficult to understand – the link between ATR and Chk1 needs to explained first
L175 AMP kinase
L176 please introduce PDAC
L180 there seems to be a reference missing
L186 tell the reader more about which genes were knocked out
L199 CRISPR editing of what? – move “Chromatin immunoprecipitation (ChIP) experiments confirmed the acetylation of the KRAS promoter histone modified by dCas9-HDAC1.” To the beginning of this paragraph
L 213 Move “Moreover, cancer cells with KRAS mutations appear to be more dependent on NPM1 expression.” To the beginning of the paragraph
L219 please introduce PD-L1 to the reader
L221 immune resistance may work better than immunotherapy response
L230-234 please explain the relevance to KRAS
L254 synthetic lethality is a concept in cells that have already one inactivated/mutated pathway therefore being dependent on the remaining second back-up pathway – this is not yet clear here
L264 please introduce “gene endoplasmic reticulum to nucleus signaling 1 (ERN1) of IRE1”
L277 FTS lack?
L282 please introduce GEA
L286-294 is difficult to understand, please revise
L299-306 please explain the link to KRAS
Figure 2 needs a legend explaining what the figure shows and what the asterisk stands for
Comments on the Quality of English Language
English needs editing to improve readability
Author Response
Comments and Suggestions for Authors
The manuscript entitled “CRISPRing KRAS: a winding road with a bright future in basic and translational cancer research” reads much better now.
Reply: We are extremely grateful for your suggestions on the content of the article, which have significantly improved the quality of the article based on your recommendations.
I have however still a few comments:
L46 please specify in how SOS1/2 interacts with RAS
Reply: Thank you for your feedback. We have included the following section in the revised manuscript. See page 2, lines 53-56.
L50 please introduce TIAM Rac1 Associated GEF 1 (TIAM1)
Reply: TIAM Rac1 Associated GEF 1 (TIAM1) is a guanine nucleotide exchange factor that is associated with the Rho GTPase Rac1. TIAM1 plays a critical role in the regulation of cell migration, adhesion, and cytoskeletal dynamics. We have added this introduction in the revised manuscript. See page 2, lines 59-60.
L58 please include that the G12C mutation is a gain-of-function mutation
L59 please introduce l AMG510 (Sotorasib) and MRTX849 (Adagrasib)
Reply: We have made changes to the above points as suggested by the reviewer.
L79-80 please include the full mutation ie G12C instead of only G12, the same applies to G13 and Q61
Reply: The citation did not mention the full mutation. Based on the reviewer's suggestion, We have provided some examples of G12, G13, and Q61 mutations.
L114 please include the full name of NSCLC
Reply: Thank you for your feedback. We have made the following revisions in the new manuscript.
L119 metabolism may work better than degradation
Reply: Thank you for your feedback. We have made the following revisions in the new manuscript.
L127 LKB1 kinase
Reply: Thank you for your feedback. We have made the following revisions in the new manuscript.
L130 STK11 needs an introduction – please explain here the link to KRAS
Reply: Please refer to the figures in the text as appropriate. It may work better to move Fig 3 to the beginning of the review after Fig 1.
Please refer to the figures in the text as appropriate. It may work better to move Fig 3 to the beginning of the review after Fig 1.
Reply: Thank you for your suggestion. We have made changes to the this points as suggested
L139 please introduce Ras-on
Reply: Thank you for your feedback. We have made the following revisions in the new manuscript.See page 5, lines 155-156.
L147 are these KRAS mutated lung tumors?
L150-161, the link to KRAS is not evident in this section, please explain
Reply: All citations are related to KRAS. For clarity, we have incorporated the relevant content into the manuscript.
L163 MAP (RAF-MEK-ERK) kinases
Reply: Thank you for your feedback. We have made the following revisions in the new manuscript.
L168 please introduce RTK signaling
Reply: Thank you for your feedback. We have made the following revisions in the new manuscript. See page 6, lines 187-188.
L170-173, this section is difficult to understand – the link between ATR and Chk1 needs to explained first
Reply: We have made changes to the this points as suggested by the reviewer.
L175 AMP kinase
Reply: We have made the following revisions in the new manuscript.
L176 please introduce PDAC
Reply: We have made the following revisions in the new manuscript. See page 6, lines 198-200.
L180 there seems to be a reference missing
Reply: The content expressed here is part of the previous research. To avoid ambiguity, we have rewritten the sentences.
L186 tell the reader more about which genes were knocked out
Reply: We have made changes to the this points as suggested by the reviewer.
L199 CRISPR editing of what? – move “Chromatin immunoprecipitation (ChIP) experiments confirmed the acetylation of the KRAS promoter histone modified by dCas9-HDAC1.” To the beginning of this paragraph
Reply: Thank you for your feedback. To avoid ambiguity, we have rewritten the sentences.
L 213 Move “Moreover, cancer cells with KRAS mutations appear to be more dependent on NPM1 expression.” To the beginning of the paragraph
Reply: Thank you for your feedback. We have made the following revisions in the new manuscript.
L219 please introduce PD-L1 to the reader
Reply: Thank you for your feedback. We have made the following revisions in the new manuscript. See page 7, lines 250-254.
L221 immune resistance may work better than immunotherapy response
Reply: Thank you for your feedback. We have made the following revisions in the new manuscript.
L230-234 please explain the relevance to KRAS
Reply: This citation is related to KRAS. For clarity, we have incorporated the relevant content into the manuscript.
L254 synthetic lethality is a concept in cells that have already one inactivated/mutated pathway therefore being dependent on the remaining second back-up pathway – this is not yet clear here
Reply: We have revised the content and references according to the reviewer's comments.
L264 please introduce “gene endoplasmic reticulum to nucleus signaling 1 (ERN1) of IRE1”
Reply: Thank you for your feedback. We have made the following revisions in the new manuscript. See page 8, lines 296-298.
L277 FTS lack?
Reply: We have revised the content according to the reviewer's comments to make it clearer.
L282 please introduce GEA
Reply: Thank you for your feedback. We have made the following revisions in the new manuscript. See page 8, lines 314-316.
L286-294 is difficult to understand, please revise
Reply: We have revised the content according to the reviewer's comments to make it clearer.
L299-306 please explain the link to KRAS
Reply: We have revised the content according to the reviewer's comments to make it clearer.
Figure 2 needs a legend explaining what the figure shows and what the asterisk stands for
Reply: Thank you for your suggestion. We have added a legend according to your comments.
Comments on the Quality of English Language:
English needs editing to improve readability
Reply: We have done a lot of English editing on the content of the manuscript, all changes have been marked.
This manuscript is a resubmission of an earlier submission. The following is a list of the peer review reports and author responses from that submission.
Round 1
Reviewer 1 Report
Comments and Suggestions for Authors
Dear Zhang Yang and colleagues,
I read your narrative review entitled “CRISPRing KRAS: a winding road with a bright future” with interest.
The review present a large number of experimental procedures mainly using the gene scissors system CRISPR to understand the biology of KRAS in malignant cells. After a brief introduction of KRAS and a short introduction of the CRISPR system, the review presents in a rather bullet point-like style the diverse experimental approaches focusing on (i) KRAS in oncogenesis, (ii) mechanisms of resistance to KRAS inhibitors, (iii) epigenetics in KRAS mutant tumours, and (iv) KRAS and synthetic lethality.
The list of references is with 115 references very extensive.
Mayor comments:
While the review offers a lot of information, the reader finds it difficult to take in the details. This is due to the absence of a clear narrative (I think the use of CRISPR alone is too weak to carry the story) and the inclusion of some sections that seem to be less related to the main theme (please see minor comments).
There are two ways to develop a stronger story line. The reader would benefit from a longer introductory paragraph where the KRAS intertwined signalling pathways - that are discussed later – are all introduced (may be with display items). This would give the reader a roadmap of what´s to come. Moreover, the less relevant sections might be deleted from the review.
Alternatively, the review is re-worked focusing on only one theme – I would like to suggest synthetic lethality – as this would give the story a focal point (that is currently absent)
Minor Comments:
Title: although the title highlights the “bright future” there is no outlook at the end of the text, which ends rather abruptly.
L55 the “characteristics of KRAS mutations” is too general. I think the current table 1 on the different CRISPR/Cas system is not needed. It would be better to list the KRAS pathways that are discussed in the review or to list some important KRAS mutations.
L76 it would work better to start section 2.2 with a short summary of how KRAS promotes tumorigenesis.
Table 2 would benefit from one additional column stating which gene had been targeted by CRISPR
L85 please start by explaining first the rational of why Matano et all introduced mutations into the tumour suppressor genes.
It would be better for the understanding when the gene would be followed by a brief functional statement in brackets e.g. TP53 (transcription factor, tumour suppressor) or LKB1 (kinase, cell proliferation)
The relevance of the following sections for the narrative is unclear: L112 -120 “Similar to LKB1, the tumour suppressor…”; L127-135 “Genomic sequencing studies have provide…”; L148-159 “Nuclear receptor binding SET…”; L172-180 “Klomp et al…”; L240-252 “Huang et al…”; L322 -327 “Kelly et al…”; L372-384 “Goodwin et al….”
L130 please explain CRC driver genes
L137 please explain RNA 00673 (LINC00673) and explain the reader why it is important to know about this
L171-172 please explain the relevance of the RAF-MEK-ERK pathway to the reader
L174 please explain PDAC
L194 please go into details about the genes needed for cell survival in the KRASg12c mutant background as this is important here
L227 NSCLC treatment in a KRAS mutant background.
L235 what is this correlation about?
L236 please introduce PD-L1
L253 please introduce the rational of the Turner et all experimental approach
L259 KRASG13D gain of function mutation
L275 please explain the relevance of the KRAS g12d/p53/KP mutant background here
L315 please explain the relevance of the ERN1 signalling pathway here
Comments on the Quality of English Language
Minor editing needed
Author Response
Dear Zhang Yang and colleagues,
I read your narrative review entitled “CRISPRing KRAS: a winding road with a bright future” with interest.
The review present a large number of experimental procedures mainly using the gene scissors system CRISPR to understand the biology of KRAS in malignant cells. After a brief introduction of KRAS and a short introduction of the CRISPR system, the review presents in a rather bullet point-like style the diverse experimental approaches focusing on (i) KRAS in oncogenesis, (ii) mechanisms of resistance to KRAS inhibitors, (iii) epigenetics in KRAS mutant tumours, and (iv) KRAS and synthetic lethality.
The list of references is with 115 references very extensive.
Reply: We thank the reviewer for the very coherent summary and positive assessment of our work.
Major comments:
While the review offers a lot of information, the reader finds it difficult to take in the details. This is due to the absence of a clear narrative (I think the use of CRISPR alone is too weak to carry the story) and the inclusion of some sections that seem to be less related to the main theme (please see minor comments).
There are two ways to develop a stronger story line. The reader would benefit from a longer introductory paragraph where the KRAS intertwined signalling pathways - that are discussed later – are all introduced (may be with display items). This would give the reader a roadmap of what´s to come. Moreover, the less relevant sections might be deleted from the review.
Alternatively, the review is re-worked focusing on only one theme – I would like to suggest synthetic lethality – as this would give the story a focal point (that is currently absent)
Reply: We thank the reviewer for the constructive comments and have revised the manuscript accordingly. In particular, we have revised the Introduction by introducing the KRAS intertwined pathways that are the focus of this review. We have also deleted the irrelevant sections. See the revised manuscript.
Minor Comments:
Title: although the title highlights the “bright future” there is no outlook at the end of the text, which ends rather abruptly.
Reply: We have changed the title and gave an outlook at the end of the text. See page 9-10, lines 348-375.
L55 the “characteristics of KRAS mutations” is too general. I think the current table 1 on the different CRISPR/Cas system is not needed. It would be better to list the KRAS pathways that are discussed in the review or to list some important KRAS mutations.
Reply: We appreciate your feedback and have deleted Table 1 in the revised manuscript as suggested by the reviewer.
L76 it would work better to start section 2.2 with a short summary of how KRAS promotes tumorigenesis.
Reply: Thank you for your valuable suggestion. We have described how KRAS mutations activate the downstream signaling and promote tumorigenesis in the Introduction and in Section 2.1.
Table 2 would benefit from one additional column stating which gene had been targeted by CRISPR
Reply: We have added a column for the gene targeted by CRISPR in the Supplementary Table 1 (original Table 2).
L85 please start by explaining first the rational of why Matano et all introduced mutations into the tumour suppressor genes.
Reply: We have described the rationale why Matano et al introduced mutations into the tumour suppressor genes (TSGs): to understand if and how a co-occurring mutation in a TSG affects KRAS-mutant cancer development. See page 3, lines 127-136.
It would be better for the understanding when the gene would be followed by a brief functional statement in brackets e.g. TP53 (transcription factor, tumour suppressor) or LKB1 (kinase, cell proliferation)
Reply: We have provided a brief functional statement for the genes described in the text.
The relevance of the following sections for the narrative is unclear: L112 -120 “Similar to LKB1, the tumour suppressor…”; L127-135 “Genomic sequencing studies have provide…”; L148-159 “Nuclear receptor binding SET…”; L172-180 “Klomp et al…”; L240-252 “Huang et al…”; L322 -327 “Kelly et al…”; L372-384 “Goodwin et al….”
Reply: Thank you for your feedback and for pointing out the sections where the relevance may be unclear. We appreciate your careful review of the manuscript and have re-written the above sections to make them clearer.
L130 please explain CRC driver genes
Reply: Thank you very much for your feedback. After considering several reviewers' comments, we have determined that this section may not be particularly relevant to our main topic. Therefore, we have decided to remove this section in the revised manuscript.
L137 please explain RNA 00673 (LINC00673) and explain the reader why it is important to know about this
Reply: Thank you for your comment regarding LINC00673. In the revised manuscript, we have provided an explanation of the significance of RNA 00673 (LINC00673) in relation to our main topic. See page 4, lines 164-168.
L171-172 please explain the relevance of the RAF-MEK-ERK pathway to the reader
Reply: Thank you for your suggestion. We have changed this part in revised manuscript. See page 5, lines 190-208.
L174 please explain PDAC
Reply: We have made changes to this point as suggested by the reviewer.
L194 please go into details about the genes needed for cell survival in the KRASg12c mutant background as this is important here
Reply: Thank you for your suggestion. According to Lou et. al (67), it is mentioned that when KRASG12C is acutely inhibited, genes that were previously less critical for the survival of cells with KRASG12C mutations may become activated, thereby supporting cell survival. They define these genes as collateral dependencies. See page 5, lines 210-213.
L227 NSCLC treatment in a KRAS mutant background.
L235 what is this correlation about?
L236 please introduce PD-L1
L253 please introduce the rational of the Turner et all experimental approach
L259 KRASG13D gain of function mutation
L275 please explain the relevance of the KRAS g12d/p53/KP mutant background here
L315 please explain the relevance of the ERN1 signalling pathway here
Reply: We have made changes to the above points as suggested by the reviewer.
Reviewer 2 Report
Comments and Suggestions for Authors
In cancers-2718828 Gong et al. review the application of CRISPR in KRAS research. This field is of interest and the review quite exhaustive. However, hereafter I list some minor comments and suggestions for possible improvement.
In general, the text is written quite sloppy with many typos and bad layout (e.g. Table 3). An English native speaker and scientist should proof-read the manuscript.
In my view, Table 1 is actually irrelevant for this review (but if it stays, very recent developments should be incorporated, e.g. Altae-Tran Science 2023). Anyway, since most of the cited studies use type II-C CRISPR/Cas9 systems, I would rather suggest changing the table toward application type, e.g. knock-out, knock-in, prime-editing, CRISPRi, CRISPRa, screens etc.
In addition, instead of a very specific Figure 1 (which is not properly referenced in the main text), I would have appreciated an overview delineating the different approaches on how CRISPR is employed in KRAS research lines (e.g. reflecting sub headers).
Degrader compounds specifically targeting mutated KRAS (now rendering it druggable) should be mentioned at some point.
Please always disclose more information on the cited model systems, like which species or cell line was used (e.g. line 243)?
Comments on the Quality of English Language
English language is okay but many typos.
Author Response
In cancers-2718828 Gong et al. review the application of CRISPR in KRAS research. This field is of interest and the review quite exhaustive. However, here after I list some minor comments and suggestions for possible improvement.
Reply: We thank the reviewer for the highly positive assessment of our work.
In general, the text is written quite sloppy with many typos and bad layout (e.g. Table 3). An English native speaker and scientist should proof-read the manuscript.
Reply: We have thoroughly improved the English to minimize errors and typos. We have also updated the Table 3(now Table 1).
In my view, Table 1 is actually irrelevant for this review (but if it stays, very recent developments should be incorporated, e.g. Altae-Tran Science 2023). Anyway, since most of the cited studies use type II-C CRISPR/Cas9 systems, I would rather suggest changing the table toward application type, e.g. knock-out, knock-in, prime-editing, CRISPRi, CRISPRa, screens etc.
Reply: We have deleted the Table 1 as suggested by this reviewer and the reviewer #1.
In addition, instead of a very specific Figure 1 (which is not properly referenced in the main text), I would have appreciated an overview delineating the different approaches on how CRISPR is employed in KRAS research lines (e.g. reflecting sub headers).
Reply: We have added a overview figure as Figure1.
Degrader compounds specifically targeting mutated KRAS (now rendering it druggable) should be mentioned at some point.
Reply: We greatly appreciate the valuable feedback from the reviewer. Following the reviewer's suggestion, we have included information about the current research status of KRAS inhibitors in the revised manuscript. See page 2, lines 53-64.
Please always disclose more information on the cited model systems, like which species or cell line was used (e.g. line 243)?
Reply: We have disclose the information as suggested by the reviewer.
Reviewer 3 Report
Comments and Suggestions for Authors
Xian Gong and co-authors present a high quality and well-written review manuscript focused on CRISPRing KRAS as a winding road with a bright future.
Authors suggest that proprietary drug development for KRAS mutant tumors has long been a challenging area of research. CRISPR technology, a highly promising gene-editing tool, is increasingly utilized in tumor studies, including those involving KRAS mutations.
Authors provide a comprehensive overview of the principles and classifications of the CRISPR system, as well as its application in KRAS research. Once deemed "undruggable" due to the strong affinity of RAS proteins for GTP and the absence of a hydrophobic "pocket" structure for drug binding.
Authors discuss the underlying principles of CRISPR technology and summarize recent research on the mechanisms of KRAS mutation, drug resistance, anti-tumor immunity, epigenetic regulatory networks, and synthetic lethality.
Authors cover such aspects as:
- Investigating the mechanism of KRAS-driven oncogenesis
- Studying the mechanisms of resistance to KRAS pathway inhibition
- Uncovering epigenetic regulatory networks in KRAS-mutant tumor develop- 201 ment
- Unravelling the crosstalk between KRAS signaling and anti-cancer immunity
- Discovering KRAS synthetic lethal interactions
Overall, the manuscript is valuable for the scientific community and should be accepted for publication after corrections are made.
======================
Other comments to authors:
1) Please check for typos throughout the manuscript.
2) Please improve figures/tables where appropriate.
3) Authors are encouraged to add one more figure.
4) Authors are encouraged to add a proper expanded Conclusion section.
5) Line 86. With regards to TP53 mutations – authors are kindly encouraged to cite the following article that describes the assessment of the stability of mutant p53 protein which is highly relevant for KRAS/p53 interplay. Since they are the two most frequently mutated genes in all human cancers.
DOI: 10.3390/life13010031
Author Response
Xian Gong and co-authors present a high quality and well-written review manuscript focused on CRISPRing KRAS as a winding road with a bright future.
Authors suggest that proprietary drug development for KRAS mutant tumors has long been a challenging area of research. CRISPR technology, a highly promising gene-editing tool, is increasingly utilized in tumor studies, including those involving KRAS mutations.
Authors provide a comprehensive overview of the principles and classifications of the CRISPR system, as well as its application in KRAS research. Once deemed "undruggable" due to the strong affinity of RAS proteins for GTP and the absence of a hydrophobic "pocket" structure for drug binding.
Authors discuss the underlying principles of CRISPR technology and summarize recent research on the mechanisms of KRAS mutation, drug resistance, anti-tumor immunity, epigenetic regulatory networks, and synthetic lethality.
Authors cover such aspects as:
- Investigating the mechanism of KRAS-driven oncogenesis
- Studying the mechanisms of resistance to KRAS pathway inhibition
- Uncovering epigenetic regulatory networks in KRAS-mutant tumor develop- 201 ment
- Unravelling the crosstalk between KRAS signaling and anti-cancer immunity
- Discovering KRAS synthetic lethal interactions
Overall, the manuscript is valuable for the scientific community and should be accepted for publication after corrections are made.
Reply: We thank the reviewer for the very coherent summary and highly positive assessment of our work.
======================
Other comments to authors:
1) Please check for typos throughout the manuscript.
Reply: We have carefully checked for the typos throughout the manuscript.
2) Please improve figures/tables where appropriate.
Reply: We have updated the figures and tables. Please see our response to reviewer #1 and #2 (above).
3) Authors are encouraged to add one more figure.
Reply: We have added an overview figure as suggested by this reviewer and reviewer #2.
4) Authors are encouraged to add a proper expanded Conclusion section.
Reply: We have elaborated a more comprehensive conclusion section.
5) Line 86. With regards to TP53 mutations – authors are kindly encouraged to cite the following article that describes the assessment of the stability of mutant p53 protein which is highly relevant for KRAS/p53 interplay. Since they are the two most frequently mutated genes in all human cancers. DOI: 10.3390/life1301003
Reply: We greatly appreciate the comments from the reviewer, but we apologize for being unable to locate the article based on the given DOI. Following your suggestion, we have found and cited a highly relevant article. See ref. 38.
Reviewer 4 Report
Comments and Suggestions for Authors
In this manuscript the authors review the underlying principles of CRISPR technology and summarizes how this technology has led to significant scientific advances in the study and treatment of KRAS-driven cancers, broadening our understanding of mechanistic insights into KRAS-driven tumor initiation, progression and resistance to therapy, such as the identification of synthetic lethal targets.
This is an interesting, detailed, comprehensive and important review focusing on the added value of the CRISP/Cas system for basic cancer research regarding KRAS mutations. However, there are some concerns/ suggestions that should be addressed:
- The introduction is completely not connected to the rest of the review and missing the main added value of this manuscript; the use of the CRISP/Cas system for basic cancer research regarding KRAS mutations.
If authors wish to summarize the CRISP/Cas technology; the information described in lines 56-74 is not adding any valuable knowledge describing in details, including Table 1, the system in prokaryotes. Instead, it is highly recommended to describe, in the text, very shortly the principals of the system and its use for gene editing in mammalian cells. In addition, Table 1 should be deleted and instead, a graphic scheme describing the system should be added. It will be much more useful and “easy” for the readers, in a review that is full of details.
- In light of the main value of this review; the Title of the manuscript, although being very ‘attractive’ for a potential reader, is missing this main point. It is highly recommended to add to the current title some of this value…..(adding for example: …with a bright future for cancer basic research).
- Both Abstracts should also emphasize the importance of the CRISP/Cas system for studying basic cancer research regarding KRAS mutations. Lines 26-27: sentence incomplete….please check and revise.
-- Section 2.1: An important paragraph, describing the use of the CRISP/Cas system for investigating and understanding KRAS-driven mutations. However, Table 2 is not “the main point” of the review, but rather the text following it; it doesn’t add any information needed to follow and understand the text; thus, table 2 should be moved to supplementary material.
- This review is describing/referring to the very complex cellular signaling pathways involved/connected to KRAS mutations. However, there is only one Figure (Figure 1) that is describing a very small portion of these pathways. Reading along the major parts of the manuscript, it’s very hard to follow all the proteins/factors mentioned….the list id endless! Thus, it is highly recommended to give in Figure 1 a much more detailed scheme of the affected signaling pathways (at least the major ones, such as MAPK, RAF-MEK-ERK and more!). Alternatively, the authors should add a partial scheme for section 2.2 and if possible for other major proteins/factors.
- When writing a review on KRAS mutations and the problem of being "undruggable" system; the impression one gets from this review is that today there are no drugs for cancers harboring KRAS mutations (see also the first sentence in the conclusions section). Therefore, one should add at this time point of research, the breakthrough discovery of covalent inhibitors as the most prevailing means of targeting the KRASG12C-specific mutation; and Sotorasib (also known as AMG510) and Adagrasib (MRTX849) that have received accelerated approval by the U.S. FDA in May 2021 and December 2022 for the treatment of non-small cell lung cancer (NSCLC) harboring KRASG12C mutations. Although this specific mutation is mentioned in the text in another context and not being discovered by the CRISP/Cas system; “old fashion” approach is still worth to mention. This information should be added to the Introduction or to the conclusion.
- The manuscript needs to undergo an extensive punctuating of the sentences and to be consistent in it’s’ rules along the whole text.
Author Response
In this manuscript the authors review the underlying principles of CRISPR technology and summarizes how this technology has led to significant scientific advances in the study and treatment of KRAS-driven cancers, broadening our understanding of mechanistic insights into KRAS-driven tumor initiation, progression and resistance to therapy, such as the identification of synthetic lethal targets.
This is an interesting, detailed, comprehensive and important review focusing on the added value of the CRISP/Cas system for basic cancer research regarding KRAS mutations. However, there are some concerns/ suggestions that should be addressed:
Reply: We thank the reviewer for the positive assessment of our work.
- The introduction is completely not connected to the rest of the review and missing the main added value of this manuscript; the use of the CRISP/Cas system for basic cancer research regarding KRAS mutations.
Reply: We have re-written the Introduction to set out the main focus of this review. See our response to reviewer #1.
If authors wish to summarize the CRISP/Cas technology; the information described in lines 56-74 is not adding any valuable knowledge describing in details, including Table 1, the system in prokaryotes. Instead, it is highly recommended to describe, in the text, very shortly the principals of the system and its use for gene editing in mammalian cells. In addition, Table 1 should be deleted and instead, a graphic scheme describing the system should be added. It will be much more useful and “easy” for the readers, in a review that is full of details.
Reply: We have deleted this part and table 1. See also our response to the other reviewers.
- In light of the main value of this review; the Title of the manuscript, although being very ‘attractive’ for a potential reader, is missing this main point. It is highly recommended to add to the current title some of this value…..(adding for example: …with a bright future for cancer basic research).
Reply: We have changed the title. See also our response to reviewer #1.
- Both Abstracts should also emphasize the importance of the CRISP/Cas system for studying basic cancer research regarding KRAS mutations. Lines 26-27: sentence incomplete….please check and revise.
Reply: We have revised the both abstracts by emphasizing the importance of CRISPR system in basic and translational cancer research.
-- Section 2.1: An important paragraph, describing the use of the CRISP/Cas system for investigating and understanding KRAS-driven mutations. However, Table 2 is not “the main point” of the review, but rather the text following it; it doesn’t add any information needed to follow and understand the text; thus, table 2 should be moved to supplementary material.
Reply: Thank you very much for the reviewer's thoughtful suggestions and feedback. We fully agree to include Table 2 (now Supplementary Table 1) in the supplementary materials.
- This review is describing/referring to the very complex cellular signaling pathways involved/connected to KRAS mutations. However, there is only one Figure (Figure 1) that is describing a very small portion of these pathways. Reading along the major parts of the manuscript, it’s very hard to follow all the proteins/factors mentioned….the list id endless! Thus, it is highly recommended to give in Figure 1 a much more detailed scheme of the affected signaling pathways (at least the major ones, such as MAPK, RAF-MEK-ERK and more!). Alternatively, the authors should add a partial scheme for section 2.2 and if possible for other major proteins/factors.
Reply: We have provided an overview figure as suggested by this reviewer and by reviewer #2, #3.
- When writing a review on KRAS mutations and the problem of being "undruggable" system; the impression one gets from this review is that today there are no drugs for cancers harboring KRAS mutations (see also the first sentence in the conclusions section). Therefore, one should add at this time point of research, the breakthrough discovery of covalent inhibitors as the most prevailing means of targeting the KRASG12C-specific mutation; and Sotorasib (also known as AMG510) and Adagrasib (MRTX849) that have received accelerated approval by the U.S. FDA in May 2021 and December 2022 for the treatment of non-small cell lung cancer (NSCLC) harboring KRASG12C mutations. Although this specific mutation is mentioned in the text in another context and not being discovered by the CRISP/Cas system; “old fashion” approach is still worth to mention. This information should be added to the Introduction or to the conclusion.
Reply: We have added the information as suggested by the reviewer.
- The manuscript needs to undergo an extensive punctuating of the sentences and to be consistent in it’s’ rules along the whole text.
Reply: We have extensively revised the manuscript and have carefully checked the grammar throughout the manuscript.